# Adapt the Data, Not the Model:
# Input-Space Adaptation for Frozen Time-Series Predictors

Omer Gotfrid [1]    Dvir Aran [1]    Barak Gahtan [1]    Alex M. Bronstein [1]

## Abstract

Clinical predictors degrade across hospitals, but regulatory and hardware constraints often forbid changing the validated model. We study domain adaptation in the locked-predictor regime for multivariate time series, where the deploying site holds target labels but the predictor's weights cannot be updated. We introduce INPUTADAPTER, a per-timestep input-space adapter trained against the frozen predictor by back-propagating the task loss using target labels, with an optional source memory bank queried via cross-attention. Across five ICU tasks (MIMIC-IV adapted to eICU and HiRID) and all five AdaTime benchmarks, IN-PUTADAPTER outperforms or ties every comparable frozen-backbone, end-to-end, and test-time adaptation baseline; it yields a +14.1 AUCPR improvement on acute kidney injury (AKI), a +10.1 mean Macro-F1 improvement over the leading test-time adaptation (TTA) method on AdaTime, and matches or exceeds natively trained target-domain models. An LSTM-trained adapter transfers zero-shot to frozen GRU and TCN predictors.

## 1. Introduction

Clinical prediction models trained at one hospital degrade when deployed at another due to differences in equipment calibration, clinical protocols, and patient populations (van de Water et al., 2024; Purushotham et al., 2018). Across the five clinical tasks on the YAIB benchmark (van de Water et al., 2024), MIMIC-trained LSTM predictors suffer cross-domain degradation of up to 14.5 AUROC points (HiRID acute kidney injury, AKI) and ~11 h MAE on HiRID length-of-stay (LoS) regression, undermining bedside reliability. Throughout, let $\mathcal{S}$ denote the labeled source domain on which the frozen predictor was trained (MIMIC-IV), and $\mathcal{T}$ the target domain where we wish to

deploy it (eICU or HiRID). Retraining on the new domain is the standard remedy, but modifying a validated predictor triggers regulatory re-certification under the FDA's Software as a Medical Device framework (U.S. Food and Drug Administration, 2021), and per-task, per-site retraining is infeasible in many clinical settings where labelled data is scarce or regulated. The same constraint arises outside healthcare: firmware-embedded models on wearable devices, industrial IoT sensors, and edge-deployed sleep-staging systems cannot be retrained per-user or per-deployment site. The result is a practical impasse: the model cannot be changed, yet it degrades on new data.

Domain adaptation (DA) methods (Ganin et al., 2016; Sun & Saenko, 2016; Wilson et al., 2020) align feature representations through a jointly trained encoder, but a frozen encoder blocks gradient-based alignment. Frozen-model alternatives from vision and NLP (Jia et al., 2022; Qiu et al., 2026) prepend learnable tokens or apply handcrafted transforms; our systematic comparison (Appendix I) shows none of them admit the sample-conditional, variable-length transformation clinical time series require, where dozens of features shift simultaneously across sparsely measured labs and continuously-monitored vitals. We propose *input-space domain adaptation*: adapt the data, not the model. Classical DA assumes a trainable encoder and unlabeled target; we address the complementary regime, where the source predictor is locked after certification and target labels are available, so task loss on the target can supervise an adapter directly. Fully unlabeled source–target adaptation is outside our scope. INPUTADAPTER encodes target windows into a latent space shared with the source domain and decodes them so that the frozen predictor receives inputs adapted to reach its decision surface. Optionally, $k$ nearest source exemplars from a memory bank are fused via cross-attention to provide additional context. Prior input-space methods apply handcrafted or unconditional transforms to single modalities such as images or univariate foundation-model inputs. IN-PUTADAPTER learns instance-conditional transformations on multivariate clinical time series with variable-length stays while keeping the predictor strictly frozen, a combination we demonstrate across five clinical tasks, four downstream architectures (LSTM, GRU, TCN, and AdaTime's 1D-CNN), and five non-clinical benchmarks.

---

[1]Technion — Israel Institute of Technology, Haifa, Israel. Correspondence to: Omer Gotfrid <omer.gotfrid@campus.technion.ac.il>.

*ICML 2026 Workshop on Foundation Models for Structured Data (FMSD)*, Seoul, South Korea, 2026.

**Contributions.**

- **Frozen predictors can be adapted through input-space learning.** When alignment gradients cannot reach a locked encoder, an adapter trained on target task labels can reshape the input instead. The predictor's weights and regulatory certification stay intact, yet adapted predictions beat every comparable frozen-backbone DA method (DANN, Deep CORAL, CoDATS, ACON), end-to-end fine-tuning, three end-to-end DA methods, and five test-time adaptation methods, across five clinical tasks and two target domains (eICU and HiRID, with MIMIC-IV as source; Appendix C, C.3). To our knowledge, no prior work addresses per-timestep frozen-predictor DA on clinical electronic health record (EHR) data.
- **Input-space adaptation reaches or exceeds natively trained models.** The adapted eICU signal reaches or exceeds the eICU-native LSTM on four of five tasks, and surpasses the in-domain MIMIC-IV-native LSTM on AKI and LoS. AKI shows the largest AUCPR gain, +14.1 points.
- **The approach is general across domains, architectures, and modalities.** On five non-clinical AdaTime benchmarks (wearable, EEG, industrial) with a frozen 1D-CNN, the adapter beats the thirteen end-to-end DA baselines on all five datasets (mean +2.6 Macro-F1; Appendix B) and exceeds the leading TTA method (TENT) by +10.1 mean Macro-F1. A reduced-capacity adapter with $\leq 1\times$ the frozen predictor's parameter count matches or exceeds the full adapter on 8/10 task–benchmark cells. An LSTM-trained adapter transfers zero-shot to frozen GRU and TCN predictors (47–86% of LSTM gain retained), so input-space adaptation is not specific to clinical time series.

## 2. Methods

### 2.1. Problem Setting

Let $\mathcal{X}_S \sim P_S$ denote source-domain clinical time series (MIMIC-IV) and $\mathcal{X}_T \sim P_T$ the target domain (eICU or HiRID); we assume the two domains share the same feature schema and inter-step sampling rate after YAIB/AdaTime preprocessing. A predictor $f_S : \mathbb{R}^{T \times F} \to \mathbb{R}^{T \times C}$, trained on $\mathcal{X}_S$, is *entirely frozen*: both its feature extractor and its final classification/regression head are non-trainable during adaptation. We learn an adapter $g_\theta : \mathbb{R}^{T \times F} \to \mathbb{R}^{T \times F}$ that maps target inputs into the source input space so that the composite system $f_S \circ g_\theta$ performs well on target labels $y_T$:

$$\mathcal{L}(\theta) = \mathcal{L}_{\text{task}}\big(f_S(g_\theta(x_T)), y_T\big) + \lambda_{\text{range}} \, \mathcal{L}_{\text{range}}\big(g_\theta(x_T)\big)$$
$$+ \lambda_{\text{tgt}} \, \mathcal{L}_{\text{tgt}}\big(\hat{y}_\theta, y_T\big) + \lambda_{\text{pred}} \, \mathcal{L}_{\text{pred}}\big(\hat{y}_\theta, y_T\big)$$
$$+ \lambda_{\text{fid}} \, \mathcal{L}_{\text{fid}} + \lambda_{\text{mmd}} \, \mathcal{L}_{\text{mmd}}\big(z_T, z_S\big), \quad (1)$$

where $\mathcal{L}_{\text{task}}$ is cross-entropy (classification) or MSE (regression) through the frozen predictor; $\mathcal{L}_{\text{range}}$ penalises outputs outside the physiological range observed in $\mathcal{X}_S$; $\mathcal{L}_{\text{tgt}}$ applies the task loss directly to the adapter output (a direct gradient that bypasses the frozen predictor) and $\mathcal{L}_{\text{pred}}$ supervises a small label-prediction head on the adapter latent; $\mathcal{L}_{\text{fid}}$ is an optional MSE fidelity penalty $\|g_\theta(x_T) - x_T\|^2$ that regularises large deviations from the unadapted input; $\mathcal{L}_{\text{mmd}}$ is an optional maximum mean discrepancy (MMD) alignment on encoder latents. The weights $\lambda_\bullet$ were selected by per-task ablation on the source-domain validation split (Appendix G). Equation (1) governs the adaptation phase; the preceding autoencoder pretrain on source data is described in §2.3. The frozen constraint reflects deployment reality: under the FDA SaMD framework, validated predictors cannot be modified post-certification (U.S. Food and Drug Administration, 2021).

### 2.2. Adapter Architecture

The INPUTADAPTER architecture comprises a shared encoder, a decoder, and, optionally, a source-domain memory bank with cross-attention fusion blocks. We treat the memory bank as a conditional component, enabled or disabled per task; the experimental breakdown of when retrieval helps is reported in Section 3.2 and Appendix G.

**Shared encoder and decoder.** A shared encoder $E_\phi$ maps both target and source windows to a latent space of dimension $d_{\text{latent}}$: $z = E_\phi(x) \in \mathbb{R}^{T \times d_{\text{latent}}}$, and a decoder $D_\psi$ maps back to input space. Each encoder and decoder layer is an *axial attention block* that applies self-attention along both temporal and feature dimensions. In clinical time series dozens of features evolve on different time scales and interact pairwise (e.g., blood pressure with vasopressor dose, creatinine with urine output), so factorising attention along the feature axis and the temporal axis lets the model capture per-feature dynamics and cross-feature interactions without flattening the full $T \times F$ sequence.

**Optional $k$-NN retrieval and memory bank.** When retrieval is enabled, the source-domain training split is encoded and segmented into temporal windows of size $W$ timesteps, each mean-pooled into a compact descriptor (Appendix B); the resulting memory bank $\mathcal{M} = \{E_\phi(x_S^j)\}_{j=1}^{|\mathcal{X}_S^{\text{train}}|}$ resides on GPU and is rebuilt every $R$ epochs as the encoder's representations evolve. For each target timestep $t$, a query is formed by mean-pooling target latents over a backward-looking window $[\max(0, t-W+1), t]$, preserving causal consistency, and the $k$ nearest source windows under Euclidean distance over latent dimensions are retrieved, with $k$ selected by grid search (Appendix B). This per-timestep design matches local physiological states rather than full trajectories.

**Cross-attention fusion.** Retrieved source context is integrated through $L$ stacked cross-attention blocks. Let $\mathcal{N}_k(z_T^t)$ denote the $k$ nearest source latents for target timestep $t$; each block computes:

$$z_{\text{cross}}^t = \text{CrossAttn}\big(Q=z_T^t,\ K=V=\mathcal{N}_k(z_T^t)\big) + z_T^t, \quad (2)$$

followed by causal self-attention and a feed-forward network, both with residual connections and pre-LayerNorm. Cross-attention allows selective, feature-level borrowing from source exemplars, so the model attends to different features from different neighbors rather than uniformly interpolating as in $k$NN-MT (Khandelwal et al., 2021). The decoder $D_\psi$ maps the fused representations back to input space:

$$\tilde{x} = D_\psi(z_{\text{cross}}) \in \mathbb{R}^{T \times F}. \quad (3)$$

The cross-attention depth $L$ is selected per task (Appendix B).

### 2.3. Training

Training follows a two-phase protocol. **Phase 1** (autoencoder pretrain): $E_\phi$ and $D_\psi$ are trained on source data $\mathcal{X}_S$ only with a reconstruction objective; ablations (Appendix G) show this phase is critical for the clinical tasks. **Phase 2** (adaptation): end-to-end training on target data through the frozen predictor with the objective of Equation (1), where $\mathcal{L}_{\text{task}}$ supplies the primary gradient through $f_S$ and the remaining terms (Equation (1)) regularize the adapter output; when retrieval is used, the memory bank is rebuilt every $R$ epochs as representations evolve.

Before adaptation, target features undergo cross-domain normalization: an affine transform matching per-feature mean and variance to source statistics, computed once from training data. Attention is causal for per-timestep tasks (AKI, Sepsis, LoS) and bidirectional for per-stay tasks (Mortality, KF), preventing information leakage across time. AdaTime datasets (HAR, HHAR, WISDM, SSC, MFD) present fixed-length windows with a single classification target at the window level, so non-causal attention does not leak labels and we use bidirectional attention throughout.

## 3. Experiments

### 3.1. Setup

**Benchmarks.** Three ICU databases via YAIB (van de Water et al., 2024): MIMIC-IV (source, 73k stays), eICU (target, 201k stays, 208 US hospitals), HiRID (second target, 34k stays). Five tasks: Mortality24 (per-stay), AKI (KDIGO≥1, 12%), Sepsis (Sepsis-3, 1.1%), LoS, kidney function (KF; serum creatinine); Appendix A. We also evaluate five AdaTime datasets (Ragab et al., 2023): HAR, HHAR, WISDM (wearable), SSC (EEG), MFD (industrial); each dataset uses its own AdaTime-pretrained 1D-CNN as the frozen predictor, with one adapter trained per source→target scenario and mean Macro-F1 reported across the ten scenarios per dataset (Appendix B).

**Baselines.** Frozen-predictor DA (unsupervised on target): DANN (Ganin et al., 2016), Deep CORAL (Sun & Saenko, 2016), CoDATS (Wilson et al., 2020), ACON (Liu et al., 2024); affine statistics-only baseline. End-to-end with target labels (supervised, like INPUTADAPTER): LSTM fine-tuning. Stress tests: three end-to-end clinical DA methods (CLUDA (Ozyurt et al., 2023), RAINCOAT (He et al., 2023), ACON; Appendix B); for AdaTime, the eleven end-to-end DA methods in Ragab et al. (2023)'s suite together with CLUDA and RAINCOAT; five TTA methods (T3A, SHOT, TENT, SAR, EATA); on the YAIB LSTM only T3A and SHOT are applicable, since TENT/SAR/EATA require BatchNorm. All headline numbers are mean±std over three seeds; $p < 0.001$ on all classification comparisons. IN-PUTADAPTER uses target labels to back-propagate the task loss through the frozen predictor (supervised target-domain adaptation, §1); the frozen-predictor DA and TTA baselines are unsupervised on the target.

### 3.2. Clinical Task Results

INPUTADAPTER surpasses all four frozen-backbone DA baselines by $1.3$–$3.7\times$, beats end-to-end fine-tuning on the three classification tasks (+0.007 to +0.032 AUROC), and reaches or exceeds the eICU-native LSTM on four of five tasks, with additional margin over MIMIC-IV-native on AKI and LoS (Table 1). AKI reaches 90.7 AUROC with $\Delta$AUCPR $+14.1$ points, the largest relative improvement across classification tasks. The Sepsis margin over DA baselines widens to $3.7\times$ (Deep CORAL $+1.67$ vs. ours $+4.91 \pm 0.42$): alignment-based methods spend gradient capacity on the domain objective, leaving little for the task signal (only $1.1\%$ of sepsis timesteps are positive).

The encode–decode path carries 84–97% of the total gain; retrieval helps Mortality and three of five AdaTime datasets but is neutral-to-negative elsewhere (Appendix G). As analysed in detail in Appendix H, adapted inputs move *away* from the source distribution (Wasserstein $+3$–$5\times$) yet predictions improve, because the adapter disrupts eICU forward-fill imputation artefacts rather than matching MIMIC's marginal statistics. On the frozen LSTM, T3A and SHOT yield no positive transfer ($\Delta$AUROC $-0.18$ to $+0.65$ for T3A, $-20.55$ to $-0.58$ for SHOT; Appendix E). On HiRID, our method's gains exceed those on eICU on AKI ($\Delta$AUROC $+7.5$ vs $+5.1$), Sepsis ($+6.9$ vs $+4.9$), and LoS ($\Delta$MAE $-12.9$ h vs $-5.4$ h), all five tasks statistically significant (Appendix C.3). The adapter is also architecture-agnostic at test time: an LSTM-trained adapter transfers zero-shot to frozen GRU and TCN predictors, retaining 47–86% of the LSTM gain despite never seeing those archi-

*Table 1.* Domain adaptation across five clinical tasks. Classification: AUROC ($\times 100$, $\uparrow$); regression: MAE ($\downarrow$). † surpasses eICU-native LSTM; ‡ surpasses MIMIC-native. ∗ marks methods supervised on target labels. Best frozen-predictor method in **bold**. Our rows report mean±std over three seeds (Appendix C.4). Fine-tune LSTM modifies the predictor; TTA rows (T3A, SHOT) run at test time (footnote §).

| Method | CLASSIFICATION (AUROC $\uparrow$) | | | REGRESSION (MAE $\downarrow$) | |
| --- | --- | --- | --- | --- | --- |
| | Mortality | AKI | Sepsis | LoS (h) | KF (mg/dL) |
| *Frozen predictor (our constraint)* | | | | | |
| Frozen baseline (source only) | 80.79 | 85.58 | 71.59 | 42.5 | 0.403 |
| Affine statistics-only | 79.79 | 84.46 | 70.87 | — | — |
| DANN | 84.38 | 88.74 | 73.23 | — | — |
| Deep CORAL | 84.53 | 88.66 | 73.26 | — | — |
| CoDATS | 84.31 | 86.84 | 71.22 | — | — |
| ACON | 84.48 | 88.62 | 71.08 | — | — |
| INPUTADAPTER∗ (MIMIC, adapted to eICU) | **85.70**[†] | **90.72**[†‡] | **76.49**[†] | **37.1**[†‡] | **0.298** |
| ±std | ±0.05 | ±0.17 | ±0.42 | ±0.3 | ±0.017 |
| Fine-tune LSTM (end-to-end)∗ | 84.61 | 90.48 | 74.59 | — | — |
| T3A (TTA, frozen)[§] | 80.61 | 82.18 | 72.24 | — | — |
| SHOT (TTA, partial-freeze)[§] | 80.21 | 65.03 | 70.49 | — | — |
| *Reference: natively trained in-domain models* | | | | | |
| *eICU-native LSTM* | *85.5* | *90.2* | *74.0* | *39.2* | *0.28* |
| *MIMIC-native LSTM* | *86.7* | *89.7* | *82.0* | *40.6* | *0.28* |

[§]TTA methods operate at test time only; SHOT partially unfreezes the LSTM feature extractor. TENT, SAR, EATA require BatchNorm layers absent from the YAIB LSTM and are omitted here; full TTA results on AdaTime are in Appendix E.

*Table 2.* AdaTime (Macro-F1 $\times 100$). Frozen-backbone adapter vs. best of thirteen end-to-end methods (eleven from Ragab et al. (2023) plus CLUDA and RAINCOAT). Mean±std over five seeds.

| Dataset | Domain | Source | Best E2E | Ours | $\Delta$ |
| --- | --- | --- | --- | --- | --- |
| HAR | Wearable | 80.0 | 93.7[a] | **94.1**±0.0 | +0.4 |
| HHAR | Wearable | 56.5 | 84.5[b] | **87.0**±0.7 | +2.5 |
| WISDM | Wearable | 50.0 | 66.3[b] | **70.3**±1.5 | +4.0 |
| SSC | EEG | 58.0 | 63.5[c] | **66.2**±0.2 | +2.7 |
| MFD | Industrial | 77.5 | 92.8[a] | **96.1**±0.1 | +3.3 |
| Mean | | 64.4 | 80.2 | **82.7** | +2.6 |

[a]DIRT-T  [b]CoTMix  [c]MMDA

tectures during training (Appendix D).

### 3.3. Generality: Non-Medical Time-Series Benchmarks

To test whether input-space adaptation generalizes beyond clinical data, we evaluate on the AdaTime benchmark (Ragab et al., 2023): a pretrained 1D-CNN is frozen; only the adapter is trained. Under AdaTime's exact training recipe, the adapter beats the thirteen end-to-end DA baselines on all five datasets (Table 2), mean +2.6 Macro-F1, and exceeds the leading TTA method (TENT) by +10.1 mean Macro-F1. A reduced-capacity variant ($\leq 1\times$ the frozen predictor's parameters) matches or exceeds the full adapter on all five datasets (best MFD: a $0.34\times$ tiny adapter at 0.9696 Macro-F1; Appendix F), showing the adapter does not rely on over-parameterisation.

## 4. Conclusion

We introduced input-space domain adaptation over frozen predictors. A single adapter outperforms every comparable baseline across five clinical tasks and five AdaTime benchmarks, exceeding the eICU-native LSTM on four of five tasks and surpassing MIMIC-IV-native on AKI and LoS, while the predictor's SaMD clearance (U.S. Food and Drug Administration, 2021) stays intact.

**Limitations.** The framework requires retrospective target labels at the deploying site, and the adapter is itself an ML component that would require its own validation in regulated deployment. Because the certified predictor receives a transformed stream, audit pipelines should log both the raw and adapted signals so decisions remain traceable to the patient's recorded physiology. Per-task ablation guidance (Appendix G) is empirically inductive and should be re-validated for novel label regimes or backbones.

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

## A. Dataset Statistics

*Table 3.* ICU databases used in this study. All data is preprocessed via the YAIB benchmark (van de Water et al., 2024) with standardized cohort definitions, feature extraction, and train/validation/test splits.

| Dataset | Role | Stays | Hosp. | Country | Source |
|---------|------|-------|-------|---------|--------|
| MIMIC-IV | Source | 73,181 | 1 | US | Academic med. ctr. |
| eICU | Target | 200,859 | 208 | US | Telehealth net. |
| HiRID | Target 2 | 33,905 | 1 | Switz. | Univ. hospital |

The 48 clinical features include vital signs (heart rate, blood pressure, $SpO_2$, temperature, respiratory rate) and laboratory values (metabolic panels, blood gases, liver enzymes, cardiac markers, coagulation studies). The 4 static features encode patient demographics (age, sex, height, weight). KF uses 192 additional generated features: cumulative minimum, maximum, mean, and count statistics for each of the 48 clinical features, which must maintain internal consistency after adaptation. LoS omits missing indicators because the YAIB LoS LSTM was trained on a 52-feature schema.

*Table 4.* Clinical prediction tasks. MI = missing indicator features (binary flags for whether a value was measured or forward-filled). Per-timestep tasks produce a prediction at every hourly timestep; per-stay tasks produce a single prediction per ICU stay.

| Task | Type | Label | Granularity | Feature Dim | Pos. Rate |
|------|------|-------|-------------|------------:|-----------|
| Mortality24 | Classification | ICU death within 24 h | Per-stay | 96 (48 + 48 MI) | $\sim$5.5% |
| AKI | Classification | KDIGO Stage $\geq$1 | Per-timestep | 100 (48 + 48 MI + 4 static) | $\sim$12% |
| Sepsis | Classification | Sepsis-3 criteria | Per-timestep | 100 (48 + 48 MI + 4 static) | $\sim$1.1% |
| LoS | Regression | Hours remaining | Per-timestep | 52 (48 + 4 static) | N/A |
| KF | Regression | Serum creatinine (mg/dL) | Per-stay | 292 (48 + 48 MI + 192 gen + 4 static) | N/A |

**Input pipeline (YAIB).** Raw EHR records contain irregular, sparse event-level measurements (lab draws at hours apart, vitals at varying frequencies). The YAIB pipeline converts these into the hourly feature matrix consumed by the frozen predictor through five steps: (i) *hourly binning*: measurements in each one-hour bin are aggregated (last value for vitals, mean for labs); (ii) *forward-fill imputation*: unmeasured features at hour $t$ inherit the last observed value, producing long flat runs for sparsely measured labs; (iii) *missing indicators* (MI): a parallel binary feature flags whether each value at hour $t$ was newly measured or forward-filled, letting the LSTM distinguish fresh from stale readings; (iv) *static broadcast*: the 4 demographic features are repeated at every timestep; (v) *cumulative statistics* (KF only): running min/max/mean/count of each vital are concatenated, adding 192 features that must remain internally consistent after adaptation. The resulting stay is a $T \times F$ matrix with $F \in \{52, 96, 100, 292\}$. Cross-domain normalization (§2.3) is applied on top. The adapter's empirical tendency to disrupt forward-fill runs on sparsely measured labs is documented in Appendix H.

## B. Implementation Details

**Physiological envelope penalty** $\mathcal{L}_{\text{range}}$. Let $[m_f, M_f]$ denote the per-feature physiological envelope on the source training corpus, set to the 0.5 and 99.5 percentiles of $X_{\text{val},S}$ in z-scored space (clipped to $[-6, 6]$ for any feature whose tail exceeds the 0.5% trim threshold). For an adapter output $\tilde{x} = g_\theta(x_T) \in \mathbb{R}^{T \times F_{\text{dyn}}}$ the range loss is the squared one-sided hinge

$$\mathcal{L}_{\text{range}} = \frac{1}{T \cdot F_{\text{dyn}}} \sum_{t,f} \left[ \max(0, m_f - \tilde{x}^{t,f})^2 + \max(0, \tilde{x}^{t,f} - M_f)^2 \right].$$

(4)

Values inside the envelope contribute zero; the squared form gives a smooth gradient at the boundary. Padded timesteps are masked out before the average. Fidelity loss is the masked MSE $\|g_\theta(x_T) - X_{\text{val},T}\|^2$ taken strictly against the dynamic-feature component $X_{\text{val},T}$ of the input tuple.

**Task-specific settings.** Temporal attention mode is set to *causal* for per-timestep tasks (AKI, Sepsis, LoS) to prevent information leakage from future timesteps, and *bidirectional*

*Table 5.* Hyperparameters for the INPUTADAPTER architecture.

| Hyperparameter | Value |
|----------------|-------|
| $d_{\text{model}}$ | 128 |
| $d_{\text{latent}}$ | 128 |
| $n_{\text{heads}}$ | 8 |
| $d_{\text{ff}}$ | 512 |
| $n_{\text{enc\_layers}}$ | 4 |
| $n_{\text{dec\_layers}}$ | 3 |
| $n_{\text{cross\_layers}}$ | 2 (Mort., Sepsis) / 3 (AKI, LoS, KF) |
| $k$ (neighbors) | 16 |
| $W$ (retrieval window) | 6 timesteps (matches the 6-hour reassessment cycle used for sepsis and AKI definitions) |
| Learning rate | $5 \times 10^{-4}$ |
| Optimizer | AdamW |
| Batch size | 16 |
| Phase 1 pretrain epochs | 15 |
| Phase 2 training epochs | 30–35 (early stopping, patience 15) |
| $\lambda_{\text{fidelity}}$ | 1.0 |
| $\lambda_{\text{range}}$ | 0.1 |
| Memory refresh ($R$) | 5 epochs |
| Dropout | 0.2 |

for per-stay tasks (Mortality, KF) where the full sequence is available at prediction time. Cross-domain normalization (an affine transform matching per-feature mean and variance to source statistics, computed once from training data) is applied to all tasks. Phase 1 trains the encoder $E_\phi$ and decoder $D_\psi$ on source data with a reconstruction objective; cross-attention blocks (when present in the architecture) receive zero context during Phase 1 and only see retrieved source context in Phase 2 once the memory bank exists. Variable-length batching is used for per-timestep tasks and disabled for per-stay tasks.

**Hardware.** Experiments used A6000-48GB (primary), RTX 3090-24GB, L40S-48GB, A100-40GB, and V100S-32GB. Cross-server variance is $\pm$0.003–0.006 AUROC across these servers.

**Model sizes.** The retrieval adapter contains 2.59M parameters (Mortality, Sepsis; $n_{\text{cross}} = 2$) to 2.88M parameters (KF; $n_{\text{cross}} = 3$). A *tiny* variant with $\leq 1\times$ the frozen predictor's parameter count uses 161K–1.43M parameters on clinical tasks and 68K–196K on AdaTime (Appendix F). Frozen

*Table 6.* End-to-end DA baselines on clinical tasks (right-padded, best hyperparameter per method). AUROC ($\times 100$, ↑). $\Delta$ vs. the frozen source-only baseline. Our Sepsis (76.49) and E2E CLUDA (76.61) are within the standard deviation across three seeds ($\pm 0.42$).

| Method | Mortality | AKI | Sepsis |
|---|---|---|---|
| Frozen source-only | 80.79 | 85.58 | 71.59 |
| E2E RAINCOAT | 81.52 | 89.55 | 76.24 |
| E2E CLUDA | 81.81 | 87.35 | 76.61 |
| E2E ACON | 82.23 | 70.79 | 75.30 |
| INPUTADAPTER (ours, frozen) | **85.70** | **90.72** | **76.49** |

LSTM baselines range from 170K (Sepsis) to 1.42M (AKI) parameters. DA baselines: DANN 350K, Deep CORAL 251K, CoDATS 428K trainable parameters.

**AdaTime configuration.** The same INPUTADAPTER architecture is used with reduced capacity: $d_{\text{model}} = 64$, $d_{\text{latent}} = 64$, $d_{\text{ff}} = 128$, $n_{\text{enc\_layers}} = 2$, $n_{\text{dec\_layers}} = 1$, $n_{\text{cross\_layers}} = 2\text{–}3$. The frozen backbone is AdaTime's standard 3-block 1D-CNN ($\sim$200K parameters). Standard AdaTime adapter ($d_{\text{model}}=64$) uses 100K–450K parameters depending on dataset; a reduced ($\leq 1\times$ predictor) variant uses 68K–196K parameters and matches or exceeds the standard adapter on all five AdaTime datasets (Appendix F). All AdaTime runs (ours and every baseline) follow the benchmark's exact recipe: 40-epoch budget, Adam with $\beta = (0.5, 0.99)$, weight decay $10^{-4}$, last-epoch model, no target-validation split.

**AdaTime end-to-end baselines.** Ragab et al. (2023) benchmark eleven state-of-the-art end-to-end DA methods: DANN, Deep CORAL, CDAN, DSAN, HoMM, MMDA, CoDATS, AdvSKM, DIRT-T, CoTMix, and SASA. We also run CLUDA (Ozyurt et al., 2023) and RAINCOAT (He et al., 2023) under the same protocol. For each dataset, Table 2 reports the best of these published methods.

**End-to-end DA on clinical tasks.** As a stress test for the frozen-predictor assumption, we also ran three end-to-end DA methods that retrain the full LSTM on each clinical task: CLUDA (Ozyurt et al., 2023), RAINCOAT (He et al., 2023), and ACON (Liu et al., 2024). All three were swept over published hyperparameters and trained under right-padded YAIB data (matching the frozen-baseline protocol). Table 6 reports the best hyperparameter per method per task. INPUTADAPTER, despite never touching the predictor, matches or exceeds the best end-to-end DA on all three tasks.

# C. Extended Results

## C.1. AUCPR Results (MIMIC → eICU)

*Table 7.* AUCPR results ($\times 100$) for classification tasks. Bootstrap 95% CIs (500 replicates).

| | Frozen Baseline | INPUTADAPTER [95% CI] | $\Delta$ |
|---|---|---|---|
| Mortality | 29.67 | $34.88 \pm 0.62$ | +5.21 |
| AKI | 56.78 | $70.95 \pm 1.56$ | +14.17 |
| Sepsis | 2.98 | $5.32 \pm 0.19$ | +2.34 |

The AKI $\Delta$AUCPR (+14.17) is nearly $3\times$ the $\Delta$AUROC (+5.14): adaptation yields larger gains on AUCPR than on AUROC, improving precision at clinically relevant operating points.

## C.2. Regression Extended Results (MIMIC → eICU)

*Table 8.* Regression results across multiple metrics. Bootstrap 95% CIs (500 replicates). MAE and RMSE: lower is better (↓); $R^2$: higher is better (↑).

| Task | Metric | Baseline | INPUTADAPTER [95% CI] | $\Delta$ |
|---|---|---|---|---|
| LoS | MAE (↓) | 0.2527 | $0.2204 \pm 0.0016$ | −0.0323 |
| LoS | RMSE (↓) | 0.3123 | 0.3004 [.3001, .3007] | −0.0119 |
| LoS | $R^2$ (↑) | 0.1130 | 0.1794 [.1782, .1806] | +0.0664 |
| KF | MAE (↓) | 0.0330 | $0.0244 \pm 0.0014$ | −0.0086 |
| KF | RMSE (↓) | 0.0600 | 0.0477 [.0451, .0503] | −0.0123 |
| KF | $R^2$ (↑) | 0.7489 | 0.8411 [.8266, .8553] | +0.0922 |

Table 8 reports our LoS configuration (without MMD) and our KF configuration (no-fidelity no-MMD with a small learning rate) across three seeds: LoS MAE $0.2204 \pm 0.0016$ denormalises to approximately 37.1 hours, a 5.4-hour reduction from the frozen baseline's 42.5 hours and 2.1 hours below the eICU-native LSTM (39.2 h); KF MAE $0.0244 \pm 0.0014$ denormalises to approximately $0.298 \pm 0.017$ mg/dL, which approaches but does not reach the eICU-native reference (0.28 mg/dL). We attribute the remaining KF gap to low kidney-function variability over short stays and to KF's reliance on 192 cumulative-statistic features that must remain internally consistent after adaptation. KF $R^2$ improves from 0.749 to 0.841, nearly doubling the explained variance in serum creatinine predictions. Dropping the fidelity loss removes fidelity gradient dominance in KF's 292-feature cumulative-statistics regime.

## C.3. MIMIC → HiRID Extended Results

HiRID results show that the adapter works across target domains with very different data collection practices (Swiss vs. US ICUs). AKI and Sepsis improvements on HiRID (+7.5 and +6.9 AUROC) exceed those on eICU (+5.1 and +4.9, matching Table 1); we hypothesise that greater domain distance provides more signal for adaptation. All five tasks are statistically significant; the smaller magnitude on KF

*Table 9.* MIMIC-IV → HiRID results. All five tasks statistically significant (three seeds).

| Task | Metric | Δ vs Frozen mean ± std (three seeds) | p |
|---|---|---|---|
| Mortality | AUROC | $+0.0589 \pm 0.0059$ | $<0.001$ |
| Mortality | AUCPR | $+0.0502 \pm 0.0084$ | $<0.001$ |
| AKI | AUROC | $+0.0753 \pm 0.0029$ | $<0.001$ |
| AKI | AUCPR | $+0.1471 \pm 0.0058$ | $<0.001$ |
| Sepsis | AUROC | $+0.0689 \pm 0.0113$ | $<0.001$ |
| Sepsis | AUCPR | $+0.0525 \pm 0.0046$ | $<0.001$ |
| LoS | MAE | $-0.0766 \pm 0.0040$ z-score ($\approx -12.9\,\mathrm{h}$) | $<0.001$ |
| KF | MAE | $-0.0017 \pm 0.0002$ | $<0.001$ |

reflects HiRID's smaller cohort and the cumulative-statistic feature regime discussed in Appendix B.

### C.4. Seed Variance (MIMIC-IV → eICU)

*Table 10.* Multi-seed variance across three seeds. Classification: mean $\Delta$AUROC ($\times 100$); regression: mean $\Delta$MAE on the task's native scale. Spread = max − min across seeds.

| Task | Adapter | Mean Δ | Spread |
|---|---|---|---|
| Mortality | classification | $+4.91$ | $0.10$ |
| AKI | classification | $+5.14$ | $0.33$ |
| Sepsis | classification | $+4.91$ | $0.72$ |
| LoS | regression | $-0.0323$ | $0.0029$ |
| KF | regression | $-0.0086$ | $0.0027$ |

AKI, Mortality, and LoS show low variance (three-seed spreads below one-third of the mean gain), well within noise. Sepsis exhibits wider spread driven by its 1.1% positive rate: a single seed can shift the AUROC by up to 0.72 points through resampling of the small positive set; this variance is inherent to the task rather than method-specific, as DA baselines show comparable variance. KF has the tightest proportional spread; the remaining gap to eICU-native is systematic rather than stochastic.

## D. Architecture-Agnostic Transfer

To test whether the adapter learns a domain-invariant input mapping rather than LSTM-specific artifacts, we apply an LSTM-trained adapter *zero-shot* to frozen GRU and TCN architectures (no retraining).

All six task–architecture combinations show positive transfer, with gains ranging from 47% to 86% of the LSTM improvement. The strongest retention occurs for TCN on Sepsis (86%) and GRU on Mortality (85%); the input-space corrections appear largely architecture-independent. This supports the interpretation that the adapter resolves distributional mismatches (e.g., forward-fill imputation artifacts) at the input level, producing features that benefit predictors

*Table 11.* Architecture-agnostic transfer: LSTM-trained adapter applied zero-shot to frozen GRU and TCN predictors. AUROC improvement ($\Delta \times 100$) over each architecture's frozen baseline. Percentage indicates fraction of LSTM gain retained.

| Task | LSTM Δ | GRU Δ (%) | TCN Δ (%) |
|---|---|---|---|
| Mortality | $+4.76$ | $+4.04$ (85%) | $+3.42$ (72%) |
| AKI | $+5.56$ | $+3.11$ (56%) | $+3.16$ (57%) |
| Sepsis | $+5.12$ | $+2.40$ (47%) | $+4.42$ (86%) |

trained on source-domain data.

## E. Test-Time Adaptation Baselines

We evaluate five TTA methods against our adapter: T3A, SHOT (Liang et al., 2020), TENT (Wang et al., 2021), SAR, and EATA. TTA methods operate at test time only (no training-time supervision) and typically require specific architectural features (BatchNorm for TENT/SAR/EATA).

**Clinical (LSTM).** T3A and SHOT are applicable to the YAIB LSTM; TENT/SAR/EATA require BatchNorm layers absent from this architecture. Three seeds, $\Delta$AUROC vs the frozen source-only baseline: T3A Mortality $-0.18$, AKI $-3.40$, Sepsis $+0.65$; SHOT Mortality $-0.58$, AKI $-20.55$, Sepsis $-1.10$. All classification tasks are net-negative or near-zero.

**AdaTime (1D-CNN).** The AdaTime backbone contains BatchNorm, enabling all five TTA methods. Mean Macro-F1 over three seeds across ten cross-domain scenarios per dataset, against the source-only baseline of 65.1: T3A 69.1, SHOT 67.7 (bimodal across datasets: HAR $-19.3$, HHAR $+17.2$), TENT 72.6, SAR 72.5, EATA 72.5. Our adapter reaches 82.7 (Macro-F1 $\Delta$: T3A $+4.0$, SHOT $+2.7$, TENT/SAR/EATA $+7.5/+7.4/+7.4$, ours $+17.6$). TENT, SAR, and EATA are tied within noise, reflecting the BatchNorm-affine regime ($\sim 448$ updated parameters).

## F. Reduced-Capacity (Tiny) Adapter

We test whether the adapter can be scaled below the frozen predictor's parameter count while retaining the gain. All tiny adapters use a reduced $d_{\mathrm{model}}$, $d_{\mathrm{latent}}$, and depth; $n_{\mathrm{cross}}$ is preserved.

Tiny adapters match or exceed the full adapter on 3/5 clinical task–benchmark cells (Mortality, LoS, KF) and exceed on all five AdaTime datasets; on MFD, a $0.34\times$ adapter (68K parameters) achieves 0.9696 Macro-F1, the best MFD result we have observed on this benchmark. This indicates that the adapter's effective capacity is substantially lower than the frozen predictor in most settings, and that the gains reported in the main body do not depend on over-parameterisation.

*Table 12.* Tiny adapter results ($\leq 1\times$ the frozen predictor's parameter count) across three seeds. Clinical tasks: $\Delta$AUROC or $\Delta$MAE. Ratio is adapter parameters divided by frozen-predictor parameters.

| Task | Tiny params | Ratio | $\Delta$ | vs full |
|---|---|---|---|---|
| Mortality | 416K | $0.86\times$ | $+0.0475 \pm 0.0013$ | matches |
| AKI | 1.43M | $1.00\times$ | $+0.0409 \pm 0.0063$ | 71% |
| Sepsis | 161K | $0.95\times$ | $+0.0194 \pm 0.0070$ | 31% |
| LoS | 1.42M | $1.06\times$ | $-0.0305 \pm 0.0003$ | 95% |
| KF | 552K | $0.98\times$ | $-0.0086 \pm 0.0004$ | matches |

*Table 13.* Tiny adapter results on AdaTime across three seeds with `pretrain_epochs=0`. Macro-F1 vs the full adapter.

| Dataset | Tiny params | Ratio | MF1 $\pm$ std | vs full |
|---|---|---|---|---|
| HAR | 196K | $0.98\times$ | $0.9438 \pm 0.000$ | $+0.31$ |
| HHAR | 196K | $0.98\times$ | $0.9093 \pm 0.003$ | $+3.89$ |
| WISDM | 196K | $0.98\times$ | $0.7595 \pm 0.016$ | $+5.68$ |
| SSC | 196K | $0.99\times$ | $0.6635 \pm 0.001$ | $\approx$ tie |
| MFD | 68K | $\mathbf{0.34\times}$ | $\mathbf{0.9696 \pm 0.002}$ | $+0.88$ |

## G. Ablation Study

We ablate the main components of the retrieval adapter. Each row modifies exactly one setting from the control configuration.

*Table 14.* Component ablation on AKI across three seeds (MIMIC-IV adapted to eICU, $\Delta$AUROC $\times 100$). Each row removes exactly one component from the full retrieval adapter ($n_{cross} = 3$). Ablation codes: C0 full; C1 no retrieval; C3 no MMD; C4 no target-task loss; C5 no fidelity; C6 no pretrain (skip Phase 1); C8 residual output.

| Code | Configuration | $\Delta$ AUROC | Spread |
|---|---|---|---|
| C0 | Full method (control) | $+4.97 \pm 0.52$ | 1.22 |
| C1 | No retrieval | $+5.14 \pm 0.17$ | 0.33 |
| C3 | No MMD alignment | $+4.99 \pm 0.18$ | 0.35 |
| C4 | No target-task loss | $+4.50 \pm 0.38$ | 0.76 |
| C5 | No fidelity loss | $+4.83 \pm 0.96$ | 1.92 |
| C6 | No pretrain (skip Phase 1) | $-1.91$ | 10.6 |
| C8 | Residual output | $+2.44 \pm 1.49$ | 2.65 |

**Key findings.** *Pretraining is essential for EHR tasks*: skipping Phase 1 collapses adaptation by up to $\sim$11 AUROC points relative to the full method (C6 row), indicating Phase 1 is required for stable transfer. On AdaTime, in contrast, top configurations use `pretrain_epochs=0`: the pre-trained CNN backbone already provides sufficient latent structure. *Retrieval is conditional, not universal*. Removing the memory bank is slightly *beneficial* on AKI (C1 $+5.14 \pm 0.17$ vs the full C0 $+4.97 \pm 0.52$) and on Sepsis ($+4.91 \pm 0.42$ vs full $+4.69 \pm 0.37$), and slightly harmful on Mortality. On AdaTime, retrieval helps HAR, SSC, and MFD by $+0.5$ to $+1.9$ Macro-F1, and hurts HHAR and WISDM by $-2.6$ to $-7.4$ Macro-F1. The encode–decode

path with the composite objective carries 84–97% of total adaptation gain. *Fidelity loss is task-dependent*: removing fidelity is catastrophic for Sepsis ($\Delta = -0.042 \pm 0.022$ at $n_{cross}=2$; $\Delta = -0.078 \pm 0.008$ at $n_{cross}=3$) and mildly harmful for LoS, but beneficial for Mortality and neutral-to-beneficial for KF. Sparse-label classification requires the fidelity anchor; dense-label classification and low-noise regression can drop it. *Residual output is brittle on dense classification*: C8 on AKI has the widest spread (2.65 points) driven by one seed at $+0.7$ vs two seeds near $+3.3$.

## H. Interpretability: Imputation vs. ACF Change

The apparent paradox in Figure 1 is that adaptation moves eICU features *away* from the MIMIC marginal distribution (Wasserstein distance rises $3.7\times$ on both AKI and Sepsis, MMD by $14\times$ and $22\times$) while task performance improves by four to six AUROC points. Under the standard domain-alignment reading of unsupervised adaptation, this should not happen: aligning marginals is the stated objective of DANN, CORAL, and their descendants, and our adapter does the opposite.

The ACF evidence in Figure 1 identifies what the adapter is actually doing. Per-feature imputation fraction, the proportion of timesteps filled by last-observation-carried-forward rather than fresh measurements, correlates strongly with the magnitude of the adaptation-induced change in lag-1 autocorrelation ($|\Delta\text{ACF}|$): Spearman $\rho=0.81$ on AKI and $\rho=0.74$ on Sepsis, both with $p < 10^{-9}$. Sparsely sampled laboratory values sit in the upper region of both panels; creatinine, bilirubin, and lactate carry imputation fractions above 0.9 and show the largest $|\Delta\text{ACF}|$. Continuously monitored vitals such as heart rate, $SpO_2$, and mean arterial pressure cluster in the lower-left region, with imputation fractions below 0.3 and near-zero $\Delta\text{ACF}$. The adapter rewrites the temporal-correlation signature of forward-filled labs and leaves genuinely sampled channels alone.

This pattern is consistent with what the frozen MIMIC-trained LSTM expects. In MIMIC, a long flat run in a lab channel encodes a specific measurement cadence typical of an academic medical center. In eICU, a visually identical flat run reflects the different ordering and telehealth-assisted cadence of community ICUs, so the same input pattern carries a different clinical meaning. The adapter disrupts the eICU forward-fill artefact so that stale channels appear stale to the predictor in the positions where staleness should matter, without attempting to match MIMIC's marginal feature statistics. Wasserstein distance increases because it measures marginal distance, which is orthogonal to the temporal operational cue the frozen LSTM was trained on.

The takeaway is that distributional alignment is a proxy, not

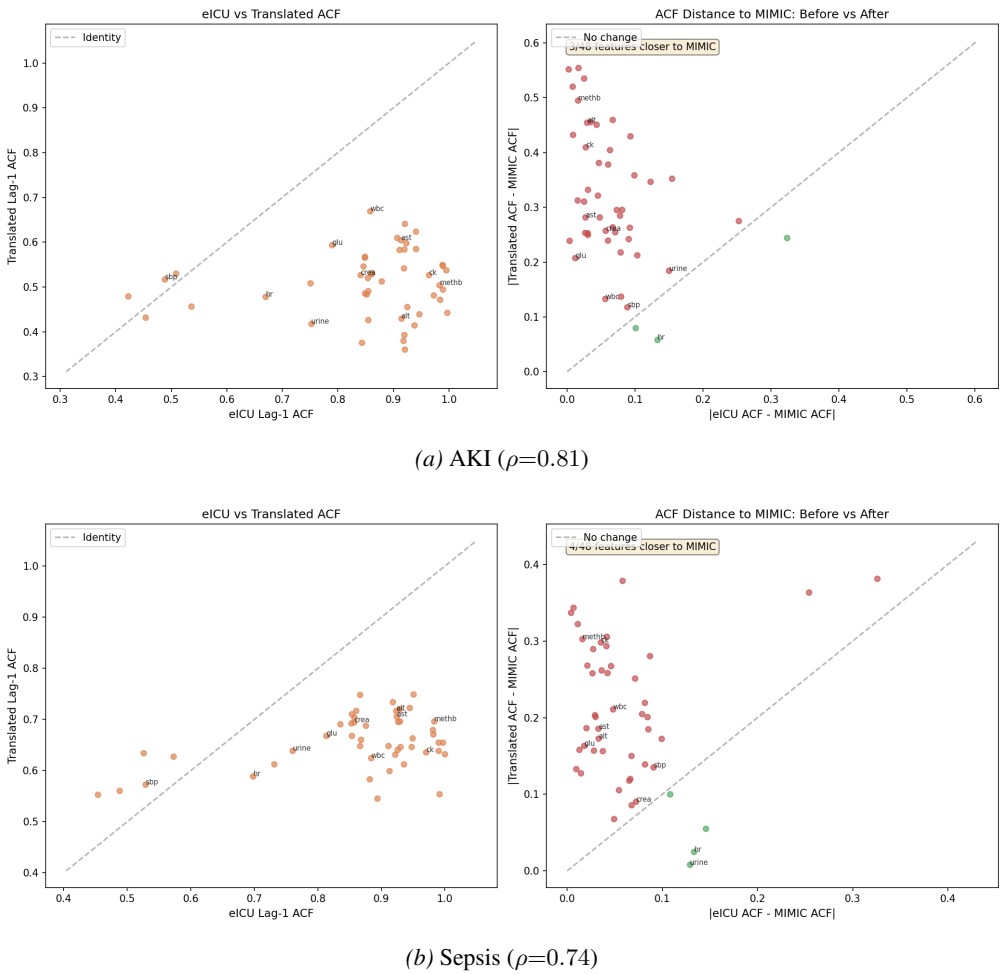

*(a)* AKI ($\rho$=0.81)

*(b)* Sepsis ($\rho$=0.74)

*Figure 1.* Feature imputation fraction vs. adaptation-induced autocorrelation change ($\Delta$ACF). Laboratory values (high imputation, upper quadrant) undergo large temporal restructuring; continuously-monitored vitals (lower left) are preserved. Spearman $\rho$ indicates the correlation between imputation fraction and ACF change, both $p < 10^{-9}$.

an objective: successful input-level adaptation restores the operational semantics the frozen predictor was trained to exploit, not the marginals of its training distribution.

## I. Systematic Method Comparison

Table 15 compares representative methods from five categories across seven dimensions that collectively define the criteria for clinical time-series domain adaptation with a frozen predictor evaluated across five clinical tasks and five non-clinical time-series benchmarks (AdaTime). To our knowledge, no prior method satisfies all seven.

Representation-space (R) and weight-space (W) methods are architecturally coupled to the model's internals and cannot be formulated without access to intermediate representations or weight tensors, violating the frozen-model constraint. Input-space methods (VPT, Time-LLM, TATO) keep the model frozen but use input-independent perturbations

(VPT, Time-LLM) or handcrafted transforms (TATO), limiting their capacity for heterogeneous clinical data where dozens of features shift simultaneously in different ways. $k$NN-MT (Khandelwal et al., 2021) is closest in spirit (frozen model, retrieval, per-timestep conditioning) but operates in output space (interpolating token distributions) rather than input space. To our knowledge, INPUTADAPTER is the only method combining a fully frozen predictor, learned input-space adaptation, sample-conditional neural transformation, source-domain retrieval, variable-length support, and evaluation across five clinical tasks.

## J. Computational Cost

Table 16 reports three-seed training time per task. Phase 1 checkpoints are reusable, amortising pretraining cost across experiments.

*Table 15.* Systematic comparison of domain adaptation and frozen-model methods across key criteria. **Model Frozen**: entire source-trained predictor weights fixed (✓), partially fixed (P), or trainable (✗). **Adapt. Space**: Input (I), Representation (R), Weight (W), or Output (O) space. **Sample-Cond.**: transformation depends on the input instance. **Retrieval**: uses a domain-specific datastore for per-instance conditioning. **Var.-Len. TS**: handles variable-length time series natively. **Clinical Tasks**: number of distinct clinical prediction tasks evaluated (0 = non-clinical only). To our knowledge, our method is the only one in Table 15 that satisfies all seven criteria simultaneously.

| Method | Model Frozen | Adapt. Space | Sample-Cond. | Retrieval | Var.-Len. TS | Clinical Tasks |
|---|---|---|---|---|---|---|
| *Clinical DA* | | | | | | |
| VRADA (Purushotham et al., 2017) | ✗ | R | ✓ | ✗ | ✓ | 1 |
| OTTEHR (OTTEHR Authors, 2025) | ✗ | R | ✗ | ✗ | ✗ | 3 |
| Mutnuri et al. (Mutnuri et al., 2024) | ✗ | W | ✓ | ✗ | ✓ | 3 |
| *General TS-DA* | | | | | | |
| CoDATS (Wilson et al., 2020) | ✗ | R | ✓ | ✗ | ✗ | 0 |
| RAINCOAT (He et al., 2023) | ✗ | R | ✓ | ✗ | ✗ | 0 |
| CLUDA (Ozyurt et al., 2023) | ✗ | R | ✓ | ✗ | ✗ | 0 |
| *Frozen-model / Input-space* | | | | | | |
| VPT (Jia et al., 2022) | ✓ | I | ✗ | ✗ | ✗ | 0 |
| Time-LLM (Jin et al., 2024) | ✓ | I | ✗ | ✗ | ✗ | 0 |
| TATO (Qiu et al., 2026) | ✓ | I | P | ✗ | ✓ | 0 |
| *Retrieval-augmented* | | | | | | |
| $k$NN-MT (Khandelwal et al., 2021) | ✓ | O | ✓ | ✓ | ✓ | 0 |
| TS-RAG (Ning et al., 2025) | ✗ | R | ✓ | ✓ | ✓ | 0 |
| *Foundation model* | | | | | | |
| ICareFM (Burger et al., 2025) | ✗ | W | ✓ | ✗ | ✓ | 4 |
| **INPUTADAPTER (Ours)** | ✓ | **I** | ✓ | ✓ | ✓ | **5** |

*Table 16.* Training time per task over three seeds on the fastest GPU available per task (A6000-48GB where possible; L40S/A100 for LoS and KF, which were not run on A6000 or 3090). Phase 1 checkpoints are reusable across experiments with matching architecture.

| Task | GPU | Phase 1 (h) | Phase 2 (h) | Total (h) |
|---|---|---|---|---|
| Mortality | A6000 | 2.5 | 5.0 | 7.5 |
| AKI | A6000 | 7.0 | 16.3 | 23.3 |
| Sepsis | 3090 | 6.8 | 13.5 | 20.3 |
| LoS | L40S | 1.4 | 18.1 | 19.5 |
| KF | A100 | 0.5 | 3.5 | 4.0 |

**Parameter counts.** Adapter: 2.59–2.88M parameters depending on task (dominated by 7 AxialBlocks at 265K each, plus 2–3 cross-attention blocks). Tiny variant (Appendix F): 161K–1.43M on clinical tasks, 68K–196K on AdaTime ($\leq 1\times$ the frozen predictor's parameter count). Frozen LSTMs: 170K (Sepsis) to 1.42M (AKI).

**Inference overhead.** At test time, the memory bank is pre-built once (~6 minutes for AKI on an A6000). Per-stay inference latency for the full adapter is ~15.5 ms (AKI, including $k$-NN retrieval from a 424K-window bank), representing ~9× overhead over the frozen LSTM alone; the tiny adapter (Appendix F) uses 0.26–0.34× the full-adapter parameter count and we did not collect dedicated latency measurements for it in this revision. The adapter is trained once per source/target dataset pair and reused across downstream architectures (Appendix D), so this overhead is paid once per deployment.

**Total compute budget.** Full development required ~2,400 GPU-hours across ~700 experiments (architecture search, ablation studies, multi-seed validation, DA baselines, HiRID experiments). Minimum reproduction of the best configuration per task (single seed, all five EHR tasks plus five AdaTime datasets) on an A6000: ~74 GPU-hours. Hardware used across the project: A6000-48GB (primary headline), RTX 3090-24GB, L40S-48GB, A100-40GB, and V100S-32GB.

**AdaTime.** Per-dataset training at a single seed: ~4 h for the full adapter across 10 cross-domain scenarios, ~1.5 h for the tiny adapter. The full three-seed benchmark across five datasets takes ~60 GPU-hours (full) or ~22 GPU-hours (tiny) on a single A6000.

