# OpenReview forum: "Adapt the Data, Not the Model: Input-Space Adaptation for Frozen Time-Series Predictors"
_ICML.cc/2026/Workshop/FMSD — FMSD @ ICML 2026 Poster_

### Official Review · Reviewer_b8Sw · 2026-05-18
**Review of Adapt the Data, Not the Model**

**Rating:** 8
**Confidence:** 4

**Review:**

## Summary

This paper proposes INPUTADAPTER, an input-space adaptation method. Instead of finetuning a validated source-domain model, the method learns an adapter that transforms target-domain inputs into source-compatible inputs. The method is evaluated on clinical transfer from MIMIC-IV to eICU/HiRID and on non-clinical AdaTime benchmarks. The paper reports strong improvements over frozen-domain-adaptation methods, end-to-end finetuning, test-time adaptation methods, and several AdaTime baselines.

---
## Strengths

1.	The paper addresses the problem of adapting deployed models when the original predictor cannot be modified due to regulatory, hardware, or deployment constraints.
2.	The core idea is simple and appealing: adapt the input data distribution rather than the model parameters. The formulation is easy to understand, and the motivation is well aligned with real-world deployment constraints.
3.	The empirical evaluation is broad. The paper tests the method on five ICU tasks, multiple frozen predictor architectures, and five AdaTime benchmarks. The reported performance gains are strong.
---
## Areas for Improvement

1.	The formulation assumes that source and target data share the same input shape and feature schema, since the adapter is defined as $g_θ:R^(T×F)→R^(T×F)$. It is uncertain whether the method would perform well if the source and target domains have different features, sampling rates, and sequence lengths.
2.	The AdaTime/TimeBench-style evaluation needs more clarification. It appears that each AdaTime benchmark uses its own source-trained frozen 1D-CNN and a task-specific adapter, rather than transferring a clinical MIMIC model to non-clinical datasets. This distinction is important because otherwise the feature semantics and input shapes would be incompatible. Moreover, AdaTime contains multiple source-to-target scenarios. The paper reports aggregate Macro-F1, but it is unclear how performance varies across different source choices. Since the frozen predictor and adapter are source-dependent, per-scenario results or variance across source-target pairs would strengthen the empirical claim.
3.	The paper sometimes refers to a “frozen predictor” and sometimes to a “frozen backbone.” It should explicitly state whether the entire predictor, including the final classification/regression head, is frozen in all experiments. If the prediction head is finetuned in any setting, this would weaken the key “adapt the data, not the model” claim.
4.	The objective contains many weighted loss terms, including task loss, range loss, auxiliary prediction losses, fidelity loss, and MMD loss. However, the paper does not fully explain how the corresponding weights are selected or how sensitive the method is to these choices. This is important because some components appear task-dependent or not consistently beneficial according to the ablation study in the appendix.
---
## Detailed Comments

1.	Please clarify the input-space assumption. Does the method require source and target domains to share the same shape after preprocessing? How would the method handle different shapes?
2.	Please clarify the AdaTime experimental protocol. For each AdaTime dataset, is there a separate source-domain 1D-CNN trained/frozen and then adapted to each target domain? Are there separate adapters per source-target scenario? If so, please provide per-source-to-target results for AdaTime, or at least variance across scenarios.
3.	Please explicitly state whether the final prediction head is frozen in all experiments. The distinction between “frozen backbone” and “frozen predictor” should be made consistent.
4.	Please report how the loss weights are selected. A sensitivity analysis over $λ_{range} $,$λ_{fid}$ ,$λ_{mmd}$,$λ_{tgt} $, and $λ_{pred}$ would help assess robustness.

---
## Justification of Score

Top 50% of accepted papers, clear accept. The paper is practically motivated, relevant to structured-data foundation-model deployment, and empirically strong. The main weaknesses are not fatal, but they affect interpretability and reproducibility, and hyperparameter sensitivity needs a clearer explanation. Overall, the contribution is interesting and likely useful for the workshop, provided the authors clarify these points.

---

### Official Review · Reviewer_Nopn · 2026-05-20
**Promising input-space adaptation method, but target-label overfitting and adapter interpretation need clarification**

**Rating:** 6
**Confidence:** 3

**Review:**

**Summary:** The paper proposes InputAdapter, a learned input-space adapter for frozen time-series predictors. Instead of updating the deployed model, the method transforms target-domain inputs so that a source-trained frozen predictor performs better. The paper evaluates the method on ICU prediction tasks and AdaTime benchmarks, reporting strong gains over several frozen-backbone, end-to-end DA, and test-time adaptation baselines.

**Strengths:** The problem setting is interesting and relevant: in some deployment regimes, adapting the input may be more practical than modifying a validated model. The empirical results are promising, especially the clinical transfer results and the breadth of comparisons. I also appreciate the ablations, reduced-capacity adapter experiments, and the attempt to analyze what the adapter changes in the input signal.

**Weaknesses:** My main concern is overfitting or target-label leakage through the adaptation procedure. The paper uses target labels to train the adapter, so the setting is supervised target-domain adaptation rather than standard unsupervised DA. This is acceptable if clearly stated, but it makes the comparison to DA and TTA baselines harder to interpret. Since the adapter is trained directly to optimize target task performance through the frozen predictor, it may learn input transformations that exploit the predictor rather than producing meaningfully adapted clinical signals.

A related concern is motivation and interpretability of the learned data adapter. The adapter is a trainable encoder-decoder placed before the frozen model, with substantial capacity in the full version. This makes it less clear whether the method is “adapting the data” in a principled sense, or effectively learning a task-specific front-end/classifier constrained only by the frozen predictor interface. The range and fidelity losses help, but I would like stronger evidence that the transformed inputs remain physiologically meaningful and that the gains are not driven by overfitting to target-domain labels.

**Suggestions:** Please clarify the exact train/validation/test protocol for target labels, including which split is used for adapter training, model selection, hyperparameter tuning, and final reporting. It would also help to include stronger controls for overfitting, such as low-label target regimes, label-shuffled controls, adapter-capacity sweeps, and comparisons to equally supervised target-domain baselines with matched parameter counts. Finally, please discuss more explicitly whether adding a learned adapter would itself require validation or certification in the clinical deployment setting.

**Justification:** I do not see an obvious fatal flaw. However, the main claims rely on a supervised target-label adaptation setting with a fairly expressive input adapter, so the results need to be interpreted carefully.